# Teaching Machine Learning in Argentina: the ClusterAI pipeline

**Martin Palazzo** [1]  **Agustin Velazquez** [1]  **Melisa Breda** [2]  **Matias Callara** [1]  **Nicolas Aguirre** [1]

## Abstract

Teaching machine learning has been a growing activity in almost any educational establishment. Despite the high availability of study materials, the Latin American region has a need of educational programs focused on machine learning. Additionally, the majority of educational materials are available only in English. In this work, we propose the ClusterAI pipeline based on a curated list of topics explained in Spanish and a collaboration with the Buenos Aires city government to open public datasets that let students apply machine learning models on real data.

## 1. Introduction

The last decade has experienced a bloom in the machine learning and data science fields propelled mainly by the improvement of new processors, data availability and new statistical learning algorithms (Goodfellow et al., 2016). Additionally, the availability of thousands of new papers, open course materials and new platforms to run and implement learning models allow the access of machine learning content to a wide audience of practitioners, researchers, and enthusiasts. Even though the availability of materials to learn machine learning has been increasing during the last years, there is a lack of content in Spanish for the Latin American region. To meet this need multiple meetings and workshops actions have been taken in the South America region such as Khipu (khipu, 2019), the Sao Paulo Advanced School on Learning from Data (spsas, 2019) and the Machine Learning Summer School 2018 (MLSS2018, 2018). These events were designed mainly for a semi-senior audience and attendees with at least 1 or 2 years of experience in the machine learning field such as Master and PhD students. Nevertheless undergraduate programs are not abundant in Argentina with the exception of an offer at Universidad de

Buenos Aires (Exactas, 2021) (FIUBA, 2017).
Additionally, we found that there are not enough open courses designed for undergraduate students with initial and basic training in algebra and statistics such as engineering, biology, social sciences, economics and design students to let them take their first steps in Machine Learning techniques and applications. In this work, we present the ClusterAI pipeline (clusterai, 2018) (`github.com/clusterai/`) held at the Universidad Tecnologica Nacional Buenos Aires. ClusterAI is a free and open-source Machine Learning program designed for last year STEM and Social sciences students. The course has multiple objectives, train on the statistical learning approach, use computational tools to run machine learning methods, use real datasets from Buenos Aires city data portal and presentation of results to a wide audience.

## 2. Course Requirements

From a computational point of view, the course does not require students to have previous programming skills. We designed a crash-course workshop that introduces students to the basics of Python and Jupyter notebooks that help students take the first steps into our proposed ClusterAI course where libraries such as Numpy, Pandas, Matplotlib and Scipy are used.
From a theoretical point of view, the course assumes the student to have a basic knowledge of linear algebra (e.g., vectors and matrices), probability (e.g., random variables and probability density functions), statistics and calculus.

### 2.1. Student profile

The ClusterAI pipeline initially started as a machine learning course in the Industrial Engineering degree at Universidad Tecnologica Nacional of Buenos Aires for last-year undergrad students which is the core audience. Nevertheless, the program has been also opened to students from other disciplines such as electronics engineering, computer engineering, biology, economics and political science. Besides any kind of heterogeneity, most of the students share a lack of formal training in any programming language nor advanced analytics.

*Equal contribution  [1]Universidad Tecnologica Nacional, Buenos Aires, Argentina [2]Subsecretaria de Politicas Publicas Basadas en la Evidencia, Gobierno de la Ciudad de Buenos Aires, Argentina. Correspondence to: Martin Palazzo <mpalazzo@frba.utn.edu.ar>.

*Proceedings of the $2^{nd}$ Teaching in Machine Learning Workshop*, PMLR, 2021.

## 2.2. ClusterAI teaching team

To allow the organization and implementation of the course three main teachers are assigned to teach in three sections of 1.5 hours each every class. Sections are theory, coding practice and case study. Additionally more than 20 mentors are voluntary participating to follow one student group for the final project.

## 3. The ClusterAI pipeline: Contents of the course and learning path

The course is divided into 7 chapters. The fundamental idea is to deliver each chapter as a workshop. The full course becomes then a sequence of workshops in which we study one specific topic at a time. The first six chapters of the course are: exploratory data analysis, supervised learning, unsupervised learning, dimensionality reduction, introduction to natural language processing and neural networks.

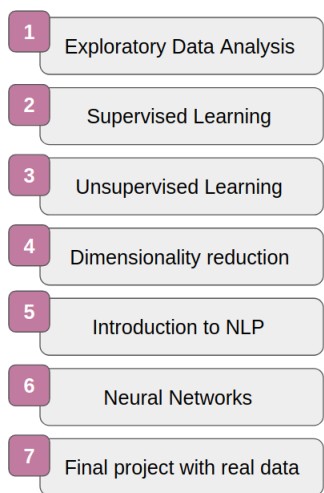

*Figure 1.* Pipeline of topics covered during the CluterAI course.

Then, chapter 7 corresponds to the final project presented by the students and described in sections 4 and 6. The bibliography used is based on papers related to each topic and books like, Elements of statistical learning (Friedman et al., 2001), Machine Learning and Pattern Recognition (Bishop, 2006), and the Deep Learning Book (Goodfellow et al., 2016) among others. The following subsections details the first six chapters of the course.

### 3.1. Exploratory Data Analysis

The exploratory data analysis (EDA) part is the initial stage of the course. The objective is to teach students how to handle, pre-process, explore, visualize and describe tabular

data. Despite Machine Learning is commonly applied to different data modalities such as Natural Language, Images, Time Series and Graphs, we decide to start with the most common and easy to work data format: tabular data.

The EDA part is composed of three sections of the course. The first three sections are more technical than theoretical since the objective is to introduce the students to the Python stack for data analysis. We assume that the learning curve of a student to get used to Numpy, Pandas, Matplotlib and Scipy can take three to four weeks. During these weeks students learn to use Jupyter Notebooks and make exploratory data analysis on tabular data, mainly CVS files.

The exploratory analysis aims to explain multiple visualizations and analytical approaches such as heatmaps, box plots, scatter plots, bar plots and histograms among others to describe visually the dataset.

To handle tabular data, Pandas and Numpy libraries are used to filter, concatenate, merge, and process data tables. Additionally, pre-processing tasks (van den Berg et al., 2006)(Jain & Bhandare, 2011) are explained since it is a core activity when dealing with open source data for the final project. All the previous concepts are explained from a theoretical point of view and simultaneously by applying with coding exercises on real data from the Buenos Aires city government open data portal such as the (GCBA, 2019).

### 3.2. Supervised Learning

Once the exploratory data analysis part is covered, the first Machine Learning concept to teach is the Supervised Learning approach. This part has three chapters about statistical learning theory and applications where classification and regression learning tasks are covered.

As an introduction to supervised learning, the concept of how samples are drawn from a d-dimensional space $\mathcal{X} \in R^d$ where $d$ is the number of dimensions or features and the multivariate approach is needed to learn from high dimensional data. Then supervised learning is explained by first introducing a hypothesis space of potential functions to be used in the learning task and the role of the loss function to pick the best one within the available space.

After defining the differences between regression and classification tasks linear and nonlinear models are explained and how these play a role in the required complexity of the model along with the variance versus bias concept and how this impact the model selection task. To deal with the model selection step cross-validation, grid search and model evaluation are studied.

The classifiers used in the course are support vector machines, k-nearest neigh boors, logistic regression and random forests. For regression tasks linear regression, polynomial regression, support vector regression and k-nearest neigh boors regression are studied.

### 3.3. Unsupervised Learning

The unsupervised learning part is based mainly on clustering and community detection methods. The idea of this part is to explain the similarity between sample vectors concept and how different measures of similarity can be used to understand if a pair of samples are similar or not. Clusters are explained and downstream analysis is studied once the segmentation of samples is obtained.
Two popular clustering methods, K-means and Hierarchical clustering, are explained and applied on Buenos Aires subway stations or house prices in Buenos Aires City datasets. Evaluation metrics such as Silhouette Index (Brun et al., 2007) and Rand Index (Hubert & Arabie, 1985) are studied to let the student decide which number of clusters is the best for a given problem.

### 3.4. Dimensionality Reduction

In the dimensionality reduction chapter linear and nonlinear unsupervised approaches are studied.
The first concept to introduce is the curse of dimensionality and how the sample to feature ratio is an important aspect to analyze before implementing a machine learning task. Additionally, a section explaining the differences between feature selection and feature extraction methods is included. The three methods studied in this chapter are Principal Component Analysis, Kernel Principal Component Analysis (kPCA) and T-distributed Stochastic Neighbor Embedding (t-SNE).
The idea for this chapter is to introduce the students on how to visualize high dimensional data in two dimensions and to reduce the complexity of a learning task due to the dimension reduction.

### 3.5. Introduction to Natural Language Processing

The idea of the NLP chapter is to make a brief introduction to other types of data formats beyond the tabular data case such as Natural language. In this chapter simple and introductory techniques such as Tokenization, Bag of Words and the TF-IDF are studied and applied on toy and real datasets. We encourage students to build their datasets with real data by taking more than 200 headlines of two newspapers from Argentina: Clarin and Pagina12. Both newspapers are known to be ambassadors of two extreme opposite political parties thus their economic headlines tend to encode signals according to two classes. They are encouraged to build the data-sets with these newspapers and learn a low dimensional representation of the economic headlines. Then supervised and unsupervised approaches are used to analyze how headers from each newspaper tend to group. Another application is the pre-processing and classification of positive and negative movie reviews, which is a popular and common teaching example for NLP tasks.

### 3.6. Neural Networks

The neural network chapter is the last one before the student project part. Neural networks are explained and trained only on tabular data.
Before studying general neural networks, the Perceptron model is presented followed by different activation functions. Then the multilayer perceptron model is presented along with different neural networks architectures such as number of hidden layers or neurons per layer. Then loss functions such as Mean Squared Error and Cross Entropy are studied in addition to the concept of local minimum in the loss landscape function. Additionally, the gradient descent and backpropagation algorithm are explained. Finally, it is explained how to improve the neural network training by regularization terms, reducing the learning rate on plateau, Dropout and Batch Normalization. With the explained concepts students are encouraged to train neural networks on simple and tabular labeled datasets such as the Wisconsin Breast Cancer or the Iris data-set and perform classification. The idea is to let students train neural networks on simple problems to understand how all the hyper-parameters affect the results. Once basic neural network implementations for supervised learning such as classification problems are introduced then the Autoencoder model is studied as an unsupervised nonlinear dimension reduction method. In this part, students benchmark low dimensional visualization tasks of the auto-encoder against the PCA and Kernel-PCA. The application case involving real data is the gene expression cancer dataset from the International Cancer Genome Consortium (Zhang et al., 2011) to let students learn low dimensional latent space of tumours and perform supervised or unsupervised downstream tasks.

## 4. Student Projects

One of the main goals for the students in this course is to develop from the ground up an applied Machine Learning based project aiming for solving real problems or getting new insights into an existing situation.
By making groups of three students, each group has the objective to pick a dataset from the Open Data Portal from the Buenos Aires city government (GCBA, 2019) to make an exploratory data analysis and implement a supervised or unsupervised learning approach to discover insights from the selected dataset. The first part of the project is the most critical since the data cleaning and pre-processing tasks are required when dealing with the open data source. All the implementations are required to be done in a Python framework as explained in section 5. Additionally, each group has assigned one alumni student serving as a mentor and helping them to get along with the objective to ensure a consistent result to be shared at the end of the course. During the course, students have different checkpoints with professors

and mentors on how to choose a profitable dataset, working strategies, best-suited algorithms and other technical issues. The final delivery is divided into 3 pillars: first, a Jupyter Notebook with the development of the project, second, a technical report explaining the root of the problem they were trying to solve, how they overcome all difficulties and the conclusions and last but not least a research poster that is presented to the public in an open event at the university where not only members of the university attend but also people from different backgrounds and disciplines.

With this methodology, students are exposed to a holistic view, from understanding the problem, building the pipeline, overcoming technical issues and presenting results in an open Data Science fair with +100 participants from different backgrounds.

## 5. Technologies used

The ClusterAI pipeline is a Python-based course. All classroom exercises and explanations are coded in Jupyter notebooks (Pérez & Granger, 2007) that serve multiple purposes: explaining and visualizing theoretical concepts with toy data, solving simple exercises with popular datasets such as the Wisconsin Breast Cancer or Iris dataset and coding implementations on real data as case studies class by class.

Sklearn and Tensorflow Keras are the main libraries to implement ML algorithms during the course as well as Pandas, Numpy (Harris et al., 2020) and Matplotlib for data wrangling and visualization are used. Additionally, all workshop material is stored in a public Github repository (clusterai, 2018) where notebooks and scripts are built and published by professors in collaboration with mentors. Each year, new content is updated and created.

For communication purposes between mentors, professors, and students the course uses Slack channels (slack, 2021). During the COVID-19 pandemic, this channel resulted useful to replace the physical room allowing students to share content and decentralize the flow of information. As additional content, there is a YouTube channel where the intention is to upload tutorials on common questions among the students, for example, installing Python and Anaconda (youtube, 2021).

## 6. Alliance with Buenos Aires Government

As explained in section 5, the last stage of the ClusterAI pipeline is to request students a machine learning application project divided into two parts: an exploratory data analysis and a machine learning application. By an alliance with the Buenos Aires City Government the datasets used are obtained from the Buenos Aires Open Data portal `data.buenosaires.gob.ar` (BA Data). The BA DATA portal is open-sourced and its datasets record approximately 60.000 downloads per month. It showcases 421 data-sets from 31 different organizations within the Buenos Aires City Government related to 12 central government initiatives: Public Administration, Culture and Tourism, Human Development, Economy and Finances, Education, Gender, Environment, Transportation, Health, Security, Urbanism and territory and COVID-19. Since December 2019, the Undersecretary of Evidence-Based Public Policies is the central team responsible for data and open data management and manages the platform, this team also suggest interesting datasets to be analyzed by the students such as transport or urban planning datasets. BA DATA is Buenos Aires City's open data portal, where public datasets are generated, saved and published. Since 2018 the ClusterAI students have done more than 50 projects using datasets from BA data. Some published projects done by the students include detection of financial behaviour using machine learning models (Weigandi, 2019), prediction of the type of vegetation (Tettamanti, 2019), commercial opportunities map (Libertun, 2019), quality air analysis via machine learning models (Cavallucci, 2019) and car robbery analysis (Carpaneto, 2019) among others.

## 7. Student feedback

Every year students complete an official survey to determine their experience during the course. Almost unanimously, the students express their agreement with the course and its organization, in addition to expressing interest in new and useful topics for the industry of the 21st century. Some anonymous responses from students are detailed below:

Student A: "The correct use of digital media that favored communication during virtuality in the pandemic was excellent."

Student B: "Updated and interesting content with resolution of cases in conjunction with demonstrations applied to the professional ecosystem"

## 8. Conclusions

In this work, we present the open ClusterAI pipeline used to teach machine learning to undergraduate students in Argentina. The proposed pipeline is composed of seven chapters where the first six are dedicated to theory and applications of multiple machine learning methods while the last chapter is focused on student projects using the Buenos Aires open data portal. Student projects include a wide range of applications using real data from city sensors. Finally, students are encouraged to present via poster sessions, technical reports and GitHub repositories the projects in order to promote open developments and to encourage the local community to use public data. Despite the ClusterAI pipeline has been designed by engineering students it has also been validated with students from other disciplines such as social science, economics and biology.

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
