# OpenReview forum: "Teaching Machine Learning in Argentina: the ClusterAI pipeline"
_ecmlpkdd.org/ECMLPKDD/2021/Workshop/TeachML — TeachML 2021_

### Official Review · Reviewer_fNBg · 2021-07-09
**A ML course for multiple background students using local data**

**Rating:** 5
**Confidence:** 4

**Review:**

In this paper, the authors present the organisation of a machine learning course for students with multiple backgrounds in a Latin American University.

Things I liked about the paper:
- The use of local data will motivate the students.
- The organisation of the course is clear and covers most topics that are currently studied in ML courses.
- Preparing materials in Spanish and open sourcing them is a useful resource for a lot of people.

In general, I liked the paper, but I think that the authors should explain the lessons that they have learned and the challenges that they have faced. In particular, I think that the paper is missing a discussion about the following points.
- It would be interesting to know the opinion of the studenst about the course. They have different backgrounds; so, how steep was the learning curve for them?
- Since students have different backgrounds, it might be interesting to select projects from their field. Have the authors thought about that?
- I would like to hear more about the projects. Open data is usually messy; so, students have to clean and organise the data. Moreover, in many cases you can find several datasets but do not know what to do with them. Was the Buenos Aires Government involved in selecting the projects?
- From the organizational point of view, how much work load means this experience for the authors? How much time did the students devote to this course? In addition, how difficult was to organise this experience at the University? How many people was
- Some of the projects might raise ethical concerns; for instance, the familiar violence estimation or the car robbery projects. Are those ethical issues explained to the students?

---

### Official Review · Reviewer_yXac · 2021-07-14
**The authors describe a project that responds to the lack of spanish language material available to teach ML, particularly for beginners. The paper provides a detailed outline for each of the seven topics covered in the course.**

**Rating:** 7
**Confidence:** 4

**Review:**

pros:
well documented outline of the content
the outline may serve as an example for others
using locally relevant data

cons:
no experience from teaching this course by instructors or student feedback etc is reported
focus us on content and data, it would be nice to have a more extensive discussion of the pedagogical approach

typos
text for Fig 1: ClusterAI course
line 153, section 3.5. Then supervised and unsupervised approaches are used
line 207, section 5 Moreover

---

### Decision · Program_Chairs · 2021-07-21

**Decision:**

Accept

**Comment:**

Congratulations! Your paper has been accepted. The reviewers agree that the paper is well written, but each reviewer offered comments on how the paper can be improved.

Camera-ready version is due August 18, 2021. As you prepare the camera ready version, please take the reviewers comments into consideration. As a minor comment, we noticed that there are a few unintentional line breaks within paragraphs. Examples include text within section 3.2, 3.3, and 3.4.

We look forward to your participation at the workshop on September 13, 2021. We invite you also to join us for the satellite event on September 08, 2021. Schedules for both the workshop and the satellite event will be forthcoming.